# The *adcA* and *lmb* Genes Play an Important Role in Drug Resistance and Full Virulence of *Streptococcus suis*

Mingzheng Peng,[a,c] Yuanyuan Xu,[c] Beibei Dou,[c] Fengming Yang,[c] Qiyun He,[c] Zewen Liu,[a] Ting Gao,[a] Wei Liu,[a] Keli Yang,[a] Rui Guo,[a] Chang Li,[a] Yongxiang Tian,[a] Danna Zhou,[a] (ID) Weicheng Bei,[b,c,d] Fangyan Yuan[a]

[a]Key Laboratory of Prevention and Control Agents for Animal Bacteriosis (Ministry of Agriculture and Rural Affairs), Hubei Provincial Key Laboratory of Animal Pathogenic Microbiology, Institute of Animal Husbandry and Veterinary, Hubei Academy of Agricultural Sciences, Wuhan, China
[b]Hubei Hongshan Laboratory, Wuhan, China
[c]State Key Laboratory of Agricultural Microbiology, College of Veterinary Medicine, Cooperative Innovation Center for Sustainable Pig Production, Huazhong Agricultural University, Wuhan, China
[d]Guangxi Yangxiang Co. Ltd., Guangxi, China

**ABSTRACT** *Streptococcus suis* is an recognized zoonotic pathogen of swine and severely threatens human health. Zinc is the second most abundant transition metal in biological systems. Here, we investigated the contribution of zinc to the drug resistance and pathogenesis of *S. suis*. We knocked out the genes of AdcACB and Lmb, two Zn-binding lipoproteins. Compared to the wild-type strain, we found that the survival rate of this double-mutant strain (Δ*adcA*Δ*lmb*) was reduced in Zinc-limited medium, but not in Zinc-supplemented medium. Additionally, phenotypic experiments showed that the Δ*adcA*Δ*lmb* strain displayed impaired adhesion to and invasion of cells, biofilm formation, and tolerance of cell envelope-targeting antibiotics. In a murine infection model, deletion of the *adcA* and *lmb* genes in *S. suis* resulted in a significant decrease in strain virulence, including survival rate, tissue bacterial load, inflammatory cytokine levels, and histopathological damage. These findings show that AdcA and Lmb are important for biofilm formation, drug resistance, and virulence in *S. suis*.

**IMPORTANCE** Transition metals are important micronutrients for bacterial growth. Zn is necessary for the catalytic activity and structural integrity of various metalloproteins involved in bacterial pathogenic processes. However, how these invaders adapt to host-imposed metal starvation and overcome nutritional immunity remains unknown. Thus, pathogenic bacteria must acquire Zn during infection in order to successfully survive and multiply. The host uses nutritional immunity to limit the uptake of Zn by the invading bacteria. The bacterium uses a set of high-affinity Zn uptake systems to overcome this host metal restriction. Here, we identified two Zn uptake transporters in *S. suis*, AdcA and Lmb, by bioinformatics analysis and found that an *adcA* and *lmb* double-mutant strain could not grow in Zn-deficient medium and was more sensitive to cell envelope-targeting antibiotics. It is worth noting that the Zn uptake system is essential for biofilm formation, drug resistance, and virulence in *S. suis*. The Zn uptake system is expected to be a target for the development of novel antimicrobial therapies.

**KEYWORDS** *Streptococcus suis*, Zn uptake system, biofilm, drug resistance, virulence

Streptococcus suis is an increasingly recognized pathogen of porcine zoonosis, which has caused huge economic losses to the pig industry (1). Of the 29 known serotypes (2), *S. suis* serotype 2 (SS2) is the most pathogenic serotype reported, and can cause a variety of severe infections, including arthritis, meningitis, endocarditis, and septicemia, and even death in swine. Infection through wounds in humans can cause bacterial encephalitis or toxic shock syndrome (3, 4). In 1998 and 2005, two large outbreaks of SS2 occurred in Jiangsu and Sichuan in China (5). These events immediately drew

Address correspondence to Fangyan Yuan, fangyanyuan12@163.com, Weicheng Bei, beiwc@mail.hzau.edu.cn, or Danna Zhou, zdn_66@yahoo.com.cn.

The authors declare no conflict of interest.

worldwide attention to *S. suis*, and antibacterial drugs are widely used for the prevention and treatment of *S. suis* infections in pigs and humans. However, antibiotic misuse promotes resistance to drugs such as fluoroquinolones, lincosamides, tetracyclines, penicillins, and macrolides (6–8). After antibiotic treatment, this microorganism has evolved many resistance mechanisms, including efflux pumps, biofilm formation, and outer membrane permeability changes (9–13). Thus, *S. suis* resistance has become prevalent in recent years, undoubtedly increasing the risk of treatment failure.

Zn is an essential cofactor required for the catalytic activity and structural integrity of various metalloproteins involved in bacterial pathophysiological processes (14–16). It has been confirmed that the Zn status of a host significantly influences its resistance to pneumococcal infection and pneumonia (17, 18). Zn stress in *Streptococcus pneumoniae* impacts numerous essential cellular processes, notably central carbon metabolism and peptidoglycan biosynthesis (19).

In streptococci, Zn acquisition is mediated by the ABC-type transporter AdcABC. In addition to AdcABC, streptococcal species encode an additional *adcA* homologue, known as *adcA*II, encoding a second Zn-binding lipoprotein (20–24). It has been reported that Zn plays an important role in the pathogenesis of Gram-positive bacteria. The Zn transporters AdcA and AdcAII in *S. pneumoniae* are essential for normal cell division and bacterial virulence (22, 25, 26). Group A *Streptococcus* (GAS) uses AdcA and Lmb to overcome the Zn limitation produced by the host, which plays an important role in the interaction between group A *Streptococcus* and its host (27–29). Group B *streptococcus* (GBS) uses AdcA, AdcAII, and Lmb to participate in Zn uptake, which is important for overcoming calprotectin-mediated stress and establishing invasive disease (30, 31). AdcABC and CntABCDF enable *Staphylococcus aureus* to compete with its host for Zn and overcome nutritional immunity, which is essential for the virulence of *S. aureus* (32). Furthermore, ZupT facilitates *Clostridioides difficile* resistance to host-mediated nutritional immunity and contributes to bacterial virulence (33). In conclusion, Zn acquisition is essential for the pathogenesis of Gram-positive bacteria. However, the Zn efflux system expressed by *S. suis* remains unknown and need to be further studied. GAS and *S. pneumoniae* adaptive responses to Zn limitation are coordinated by the Zn-sensing transcription regulator adhesion competence repressor (AdcR) (21, 24, 34). AdcR belongs to the multiple-antibiotic resistance family of regulators and mediates Zn-dependent transcriptional regulation of genes involved in Zn scavenging, sparing, and acquisition during Zn limitation (24, 35).

On the other hand, the mammalian host can restrict the availability of Zn to invading pathogens by producing small molecules and proteins that tightly bind to metals, an active process called nutritional immunity (36). For example, calprotectin (CP), a Mn/Zn-sequestering protein of the S100 family (37) which is produced abundantly by neutrophils and found at high concentrations within inflammatory sites during infection, restricts the bioavailability of Zn in the host (27, 38, 39). Harnessing the antimicrobial activity of exogenous Zn has been suggested as a means to potentiate the efficacy of antibiotic treatment. Zn intoxication manifests as reduced resistance to various antibiotic classes in *S. pneumoniae*. The innate immune response manipulates the chemistry of niches such as the lungs in the context of pneumococcal infection (40) and exploits the antimicrobial activity of Zn (41).

Biofilms help bacteria escape the killing effects of antibiotics and the host immune system. In *S. suis*, the biofilm plays a key role in meningitis (42). The ability of *S. suis* to form biofilms in a host causes persistent infections which are difficult to eradicate with antibiotics (43) and inhibit the formation of extracellular neutrophil traps (44). Zinc has crucial structural and catalytic roles in the proteome of all organisms. The immune response generates a variety of antimicrobial agents to control infection, including zinc stress. The effect of zinc on biofilm formation and drug resistance in *S. suis* is unknown. This study aimed to identify the role of Zn acquisition systems in *S. suis*.

We determined that the expression levels of *adcA* and *lmb* in the wild-type strain increased significantly in Zn-restricted medium. Next, we found that Δ*adcA*Δ*lmb* could not grow in Zn-restricted medium but could grow in chemically defined medium

(CDM) supplemented with Zn. AdcA and Lmb contributed to biofilm formation, adhesion to and invasion of cells, and full virulence in the mouse model. In addition, the Δ*adcA*Δ*lmb* strain downregulated the expression of adhesion-related genes and exhibited more sensitivity to cell wall-targeting drugs. Collectively, these results suggest that AdcA and Lmb have overlapping roles in Zn acquisition and bacterial virulence. The bacterial Zn uptake system plays a very important role in *S. suis* pathogenicity and is expected to be a target for the development of novel antimicrobial therapies.

## RESULTS

**Bioinformatics analysis of AdcA and Lmb.** Based on analysis of the *S. suis* genome and protein sequence alignment, we found that the homology of *S. suis* SSUSC84_RS00725 protein with the Zn transport protein AdcA in *S. pneumoniae*, *Streptococcus pyogenes*, and *Streptococcus agalactiae* was 71%, 60%, and 57%, respectively. The *S. suis* SSUSC84_RS01665 gene encodes a laminin-binding protein and has 63%, 58%, and 56% homology with the Zn transport proteins in *S. pneumoniae*, *S. pyogenes*, and *S. agalactiae*, respectively.

**Construction of the deletion mutant and complementation strain.** According to the designed internal and external primers, the wild-type, deletion mutant, and complementary strains were verified by PCR and reverse transcription PCR (RT-PCR) amplification. The results showed that Δ*adcA*, CΔ*adcA*, Δ*lmb*, CΔ*lmb*, and Δ*adcA*Δ*lmb* had been successfully obtained (Fig. 1). The sequence accuracy of each strain was verified by DNA sequencing (data not shown).

**AdcA and Lmb promote growth of *S. suis* in Zn-restricted environments.** We found there was no difference in growth between the Δ*adcA*Δ*lmb* and WT strains in tryptic soy broth (TSB). To explore the role of AdcA and Lmb of *S. suis* in Zn-restricted environments, we tested the transcription levels of *adcA* and *lmb* in CDM with or without Zn. As shown in Fig. 2, the transcription levels of *adcA* and *lmb* increased significantly in CDM without Zn compared to medium with Zn. Furthermore, the transcription levels of *adcA* and *lmb* decreased as the concentration of Zn in CDM increased. To uncover the role of AdcA and Lmb in Zn uptake, we generated Δ*adcA*, Δ*lmb*, and Δ*adcA*Δ*lmb* strains using a markerless system. Upon genetic confirmation that all gene deletions had occurred as planned, the WT and mutant strains were grown in CDM without Zn. In the absence or presence of low concentrations of Zn (0, 0.0001, 0.001, and 0.01 mM), Δ*adcA*Δ*lmb* showed a significant growth defect, while there was no significant difference in growth between the single-mutant and wild-type strain (Fig. 3A to D). All strains showed growth inhibition at high Zn concentrations (data not shown). The main function of the metal substrate-binding protein is to transport metal ions. Collectively, these results demonstrated that AdcA and Lmb both contribute to the growth of *S. suis* in Zn-restricted environments.

**The absence of AdcA and Lmb affects bacterial structure and integrity.** Bacterial cells of the WT strain in CDM with or without 0.01 mM Zn were coccoid or ovoid (Fig. 4A and C), usually present as single cells, in pairs, or in short chains. However, the Δ*adcA*Δ*lmb* strain in CDM without Zn was greatly affected compared to that in CDM with Zn (Fig. 4B and D). The Δ*adcA*Δ*lmb* strain displayed lots of cellular damage, such as plasmolysis, inhomogeneous electron density, and uneven capsule thickness. The irregular cell shape was due to an indistinct cell wall and a collapsed cell membrane.

**AdcA and Lmb contribute to invasion of host cells and survival in pig blood.** In *S. pneumoniae*, *adcA*II contributes to invasion (45). The blood-brain barrier (BBB) protects the brain from bacteria in the bloodstream (46). *S. suis* can breach the BBB and cause meningitis; its first step is adhering to brain microvascular endothelial cells (BMEC) (47). Our laboratory has previously found that *S. suis* can adhere to human BMEC (hBMEC) and invade hep-2 cells (48). Adhesion and invasion assays were conducted in accordance with previously described methods (48). To analyze whether AdcA and Lmb also play an important role in adherence to and invasion of host cells, we tested the adhesion ability of the WT and mutant strains to hBMEC. As shown in Fig. 5A, the capacity of the Δ*adcA*Δ*lmb* strain to adhere to hBMEC was significantly

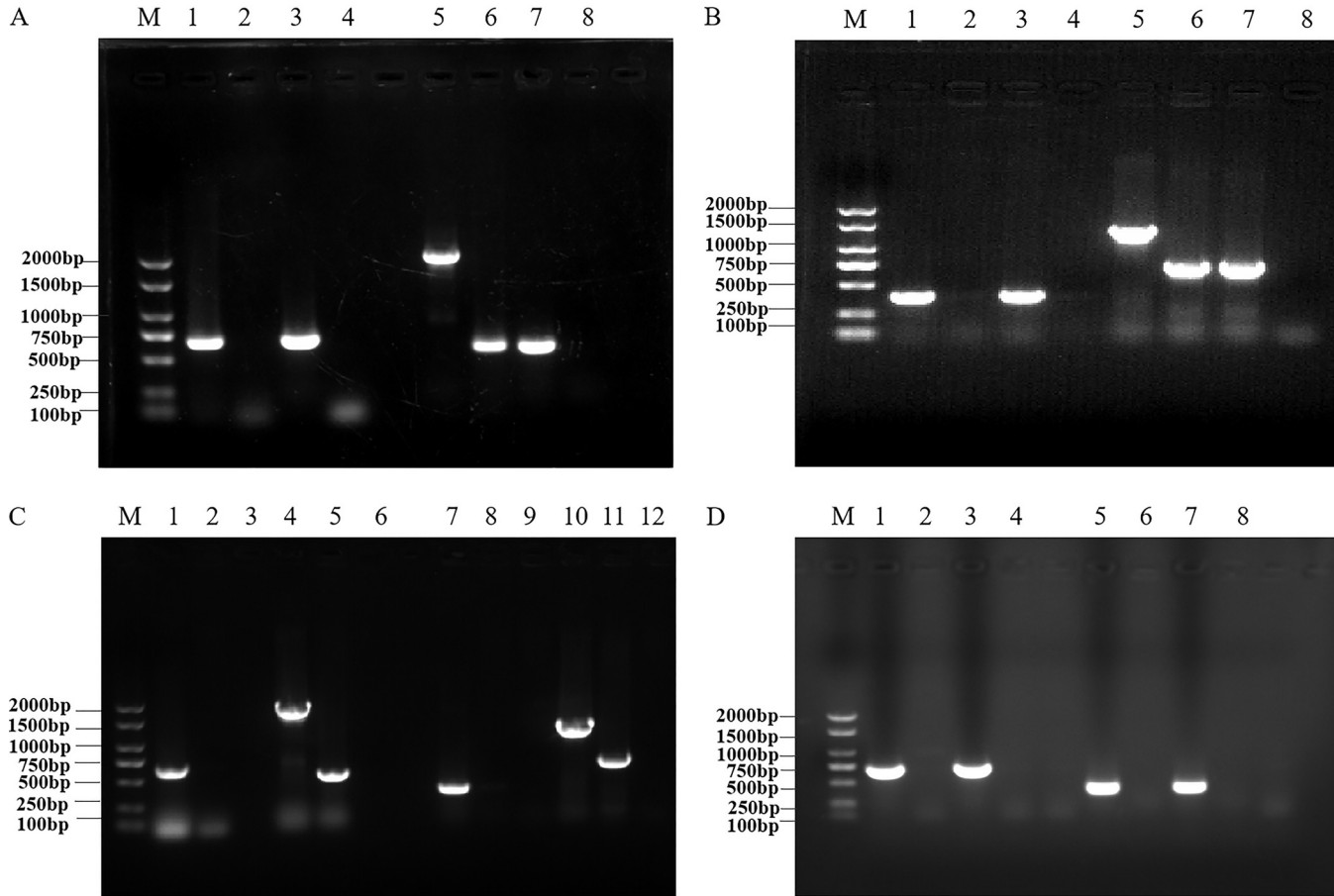

**FIG 1** Identification of the mutant and complementation strains. (A) Confirmation of ΔadcA and CΔadcA strains by PCR. Lanes 1 to 3 represent the amplification of SC19, ΔadcA, and CΔadcA using the primer pair In-adcA-F and In-adcA-R. Lanes 5 to 7 represent the amplification of SC19, ΔadcA, and CΔadcA using the primer pair Out-adcA-F and Out-adcA-R. Lanes 4 and 8 represent the negative control. (B) Confirmation of Δlmb and CΔlmb by PCR. Lanes 1 to 3 represent the amplification of SC19, Δlmb, and CΔlmb using the primer pair In-lmb-F and In-lmb-R. Lanes 5 to 7 represent the amplification of SC19, Δlmb, and CΔlmb using the primer pair Out-lmb-F and Out-lmb-R. Lanes 4 and 8 represent the negative control. (C) Confirmation of ΔadcAΔlmb by PCR. Lanes 1 to 5 represent the amplification of SC19 and ΔadcAΔlmb using the primer pair In-adcA-F and In-adcA-R. Lanes 4 and 5 represent the amplification of SC19 and ΔadcAΔlmb using the primer pair Out-adcA-F and Out-adcA-R. Lanes 7 and 8 represent the amplification of SC19 and ΔadcAΔlmb using the primer pair Out-adcA-F and Out-adcA-R. Lanes 10 to 11 represent the amplification of SC19 and ΔadcAΔlmb using the primer pair Out-lmb-F and Out-lmb-R. Lanes 3, 6, 9, and 12 represent the negative control. (D) Confirmation of ΔadcA, Δlmb, and ΔadcAΔlmb by RT-PCR. Lanes 1 to 4 represent the amplification of cDNA from SC19, ΔadcA, Δlmb, and ΔadcAΔlmb using primer pair In-adcA-F and In-adcA-R. Lanes 5 to 8 represent the amplification of cDNA from SC19, ΔadcA, Δlmb, and ΔadcAΔlmb using primer pair In-lmb-F and In-lmb-R.

lower than that of the WT strain. Furthermore, we also tested the invasion ability of WT and mutant strains to hep-2 cells. As shown in Fig. 5B, the abilities of the Δlmb and ΔadcAΔlmb to invade hep-2 cells were significantly lower than that of the WT strain. We also performed the same experiments with PK-15 cells and obtained similar results (data not shown). In addition, we found that the growth and proliferation of ΔadcAΔlmb decreased significantly in pig blood compared with that of the WT strain. These data reveal that *adcA* and *lmb* are necessary for mediating bacterial adhesion to and invasion of host cells and bacterial survival in pig blood (Fig. 5C).

**AdcA and Lmb contribute to biofilm formation and drug resistance.** Biofilms help bacteria escape the killing effects of antibiotics and the host immune system. In *S. suis*, the biofilm plays a key role in meningitis. We tested whether the *adcA* and *lmb* genes in *S. suis* modulate biofilm formation on polystyrene surfaces. The biofilm formation of the ΔadcA, Δlmb, and ΔadcAΔlmb strains was significantly lower than that of the WT strain in quantitative experiments (Fig. 6A and B). Next, we tested the relative expression of adhesion genes by the WT and ΔadcAΔlmb strains. Among these, the expression levels of *gor*, *gapdh*, *gdh*, *perR*, *ccpA*, *fbps*, *srtA*, and *gdpP* in the ΔadcAΔlmb strain were significantly lower than those in the WT strain (Fig. 6C). In addition, the

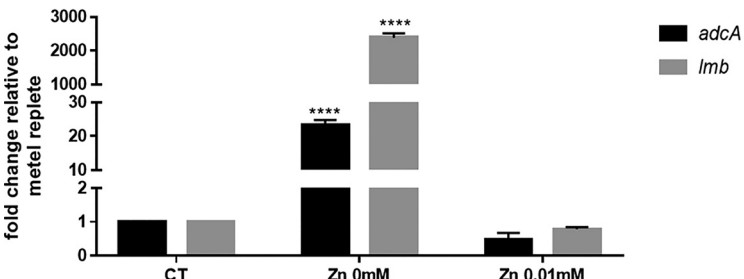

**FIG 2** The transcription levels of *adcA* and *lmb* of *Streptococcus suis* in chemically defined medium (CDM) with or without Zn supplementation. *S. suis* was incubated with different concentrations of Zn in CDM for 3 h, after which bacterial RNA was isolated and purified, and expression levels of *adcA* and *lmb* were measured by reverse transcription-quantitative PCR (RT-qPCR). The control group was incubated in tryptic soy broth (TSB) for 3h. The relative expression level of each gene was normalized to that of the housekeeping gene 16S rRNA. Results from three independent assays are expressed as means ± standard deviation (SD). Statistical analyses were performed using a two-tailed unpaired *t* test.

disruption of AdcA and Lmb lowers the tolerance of *S. suis* to cell envelope-targeting antibiotics (Fig. 7A to D); the specific mechanism for this needs to be further studied.

**Zn is critical for *S. suis* virulence during infection.** Zn transporters, such as *adcA* and *adcA*II in *S. pneumoniae* (22); *adcA* and *lmb* in *S. pyogenes* (29); *adcA*, *adcA*II, and *lmb* in *S. agalactiae* (30); *adcA* and *cntA* in *S. aureus* (32); and *zupT* in *Clostridium difficile* (33); have been reported to contribute to streptococcal virulence. To determine whether AdcA and Lmb in *S. suis* are also relevant to virulence, we tested the virulence of WT, Δ*adcA*, CΔ*adcA*, Δ*lmb*, CΔ*lmb*, and Δ*adcA*Δ*lmb* strains in a mouse model. As shown in Fig. 8A, the survival rate of the Δ*adcA*Δ*lmb* group was 100%, with mice showing no obvious clinical symptoms. Meanwhile, the survival rate of the WT group was 20%, and the surviving mice were thin and depressed. The survival rate of the Δ*adcA*Δ*lmb* group was significantly higher than that of the WT group (Fig. 8A). Furthermore, we tested the colonization ability of the WT and mutant strains. Forty-eight hours postinfection, we calculated the bacterial counts of the strains in mice. The Δ*adcA*Δ*lmb* strain was barely recovered from the heart, liver, spleen, lung, kidney, and brain of infected mice, whereas significant numbers of the

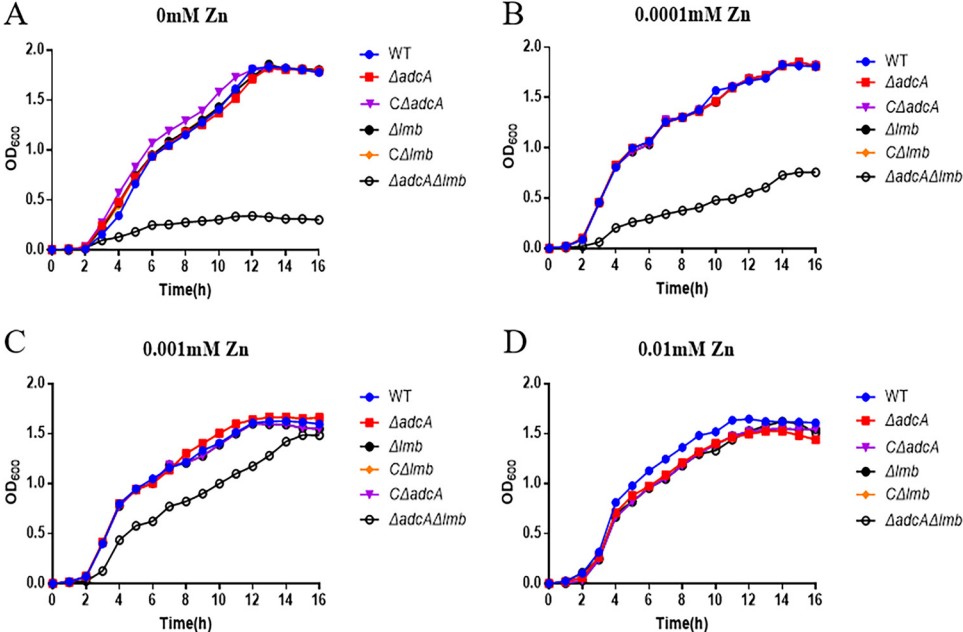

**FIG 3** AdcA and Lmb play a important role in Zn uptake for *S. suis* in CDM. Growth curves of the wild-type (WT), Δ*adcA*, Δ*lmb*, and Δ*adcA*Δ*lmb* strains in the absence (A) and presence of 0.0001 (B), 0.001 (C), and 0.01 mM Zn (D). Data points represent the average of three biological replicates.

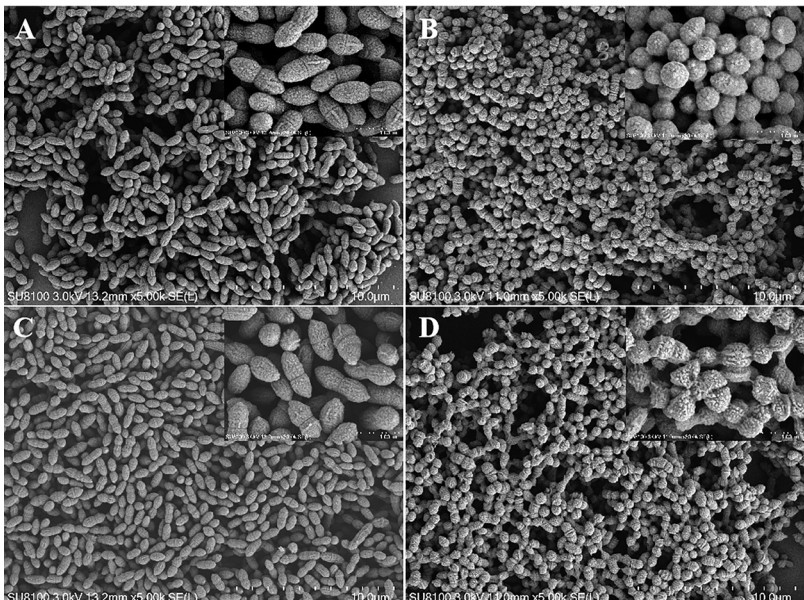

**FIG 4** Effect of Zn on the bacterial morphology of wild-type and ΔadcAΔlmb strains. Scanning electron micrographs of the WT strain in CDM with 0.01 mM Zn (A) or without Zn (C) and the ΔadcAΔlmb strain in CDM with 0.01 mM Zn (B) or without Zn (D). Scale bars = 10 μm.

WT strain were found (Fig. 8D to F). The absence of *adcA* and *lmb* significantly reduced the colonization ability of *S. suis* in mice. These results reveal that the Zn transporters AdcA and Lmb play an important role in the virulence of *S. suis*. To further explore the proinflammatory ability of the WT and mutant strains, we determined the bacterial load and cytokine level in the blood of mice at different time points post-infection. The levels of interleukin (IL)-6, IL-1$\beta$, and tumor necrosis factor $\alpha$ (TNF-$\alpha$) after infection with ΔadcAΔlmb at 9 and 12 h were significantly lower than those after WT infection (Fig. 8I to K). Collectively, the ΔadcAΔlmb strain was more easily cleared by the host compared with the WT strain at 9 and 12 h post-infection (Fig. 8B and C). AdcA and Lmb have no influence on inflammatory cytokine production during the early stage of infection. To further evaluate the role of AdcA and Lmb in *S. suis* pathogenicity, liver, brain, spleen, and lung tissues of infected mice were collected for hematoxylin and eosin staining and pathological observation. As shown in Fig. 9, the tissues collected from the WT mice showed various pathological changes. The lung tissues (Fig. 9, lane 3) exhibited telangiectasia, bleeding (red arrows), alveolar wall thickening (yellow arrows), alveolar atrophy (blue arrows) and inflammatory cell infiltration (black arrows). Liver tissues (Fig. 9, lane 1) exhibited mild inflammatory cell infiltration (black arrows) and punctate necrosis (red arrows). In the spleen (Fig. 9, lane 2), the white pulp was

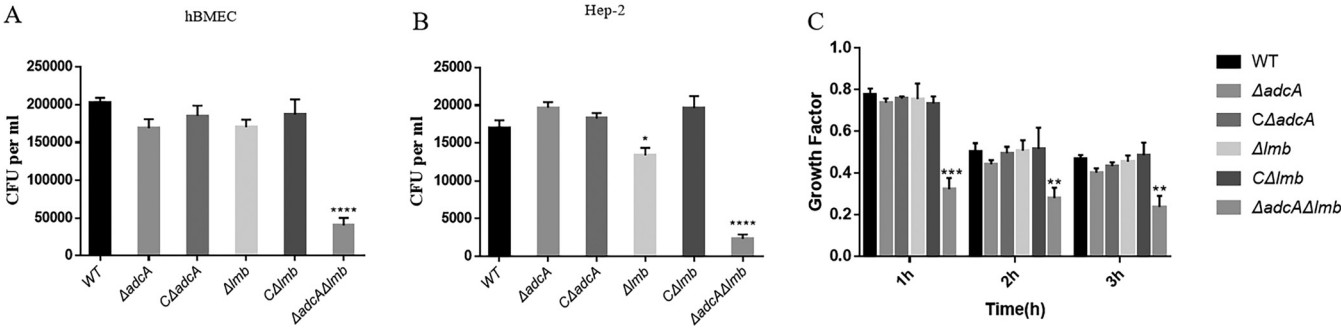

**FIG 5** Cell assays and pig blood killing assay. (A) Ability of *S. suis* strains to adhere to human brain microvascular endothelial cells (hBMEC). (B) Ability of *S. suis* strains to invade hep-2 cells. (C) Growth and proliferation of the WT, ΔadcA, CΔadcA, Δlmb, CΔlmb, and ΔadcAΔlmb strains in pig blood. *S. suis* was incubated with pig blood for 1, 2, and 3 h, respectively. Data points represent the average of three biological replicates. Statistical analyses were performed using a two-tailed unpaired *t* test.

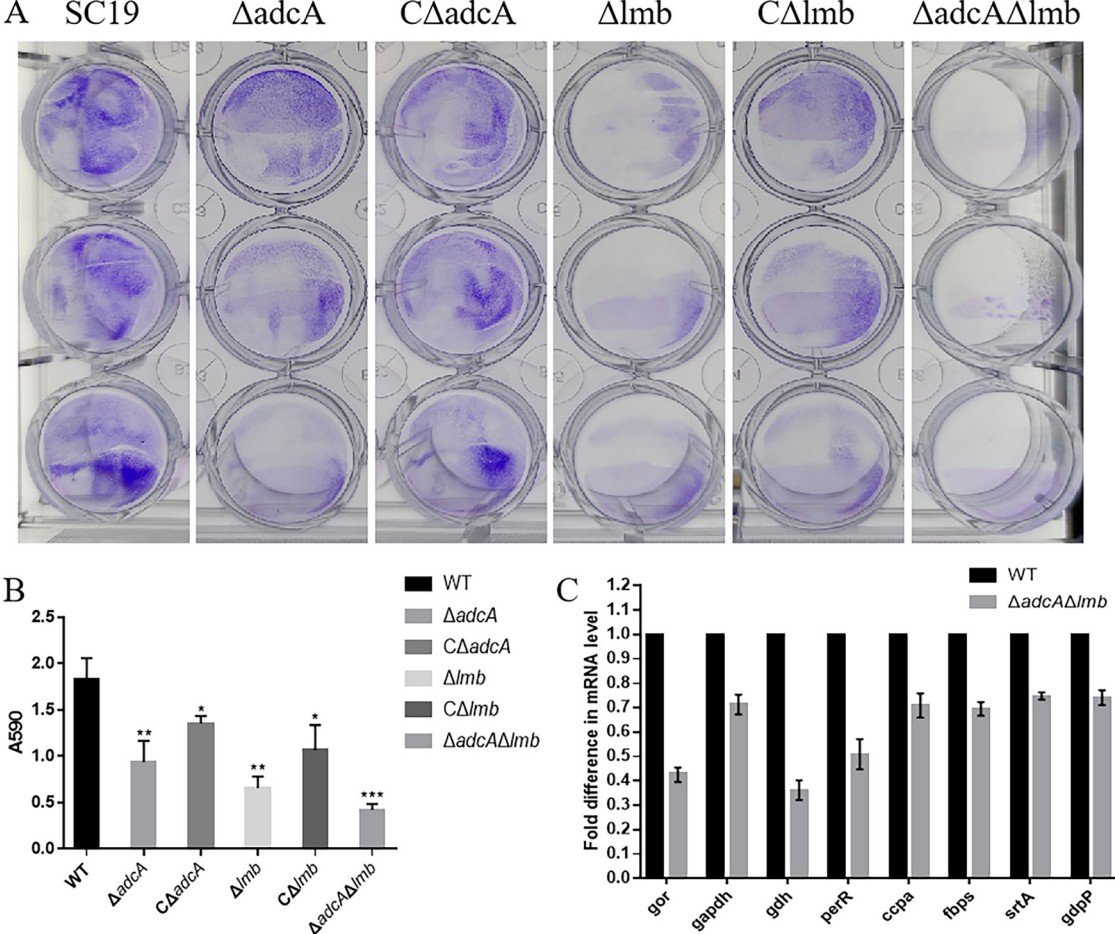

**FIG 6** AdcA and Lmb contribute to biofilm formation in *S. suis* by downregulating the expression of adhesion genes. The WT, Δ*adcA*, CΔ*adcA*, Δ*lmb*, CΔ*lmb*, and Δ*adcA*Δ*lmb* strains were incubated in TSB for 24 h in 37°C. (A) Crystal violet staining of biofilms was performed in a 24-well microplate. (B) Absorbance was read at 590 nm; data points represent the average of three biological replicates. (C) Relative expression of adhesion genes by the WT and Δ*adcA*Δ*lmb* strains. Gene expression level for the WT strain was set at 100%. Gene expression levels for the Δ*adcA*Δ*lmb* strain are given relative to those of the WT strain. Results from three independent assays are expressed as means ± SD. Statistical analyses were performed using a two-tailed unpaired *t* test.

disturbed and indistinct from the red pulp (green arrows) and there was bleeding in the splenic parenchyma (red arrow; blue arrows indicate the splenic trabeculae). In addition, the changes in brain tissue (Fig. 9, lane 4) included the disappearance of pyramidal cells in the CA3 region of the hippocampus, neurofibrillary tangles in a few neurons (yellow arrows), neural cell degeneration in the dentate gyrus (DG) region (blue arrows), and heterotypic cell infiltration in the DG (red arrows). However, only slight inflammatory cell exudation was observed in the tissues collected from the Δ*adcA*Δ*lmb* mice. No other obvious abnormalities were seen (Fig. 9).

## DISCUSSION

During infection, nutritional immunity severely restricts the bioavailability of the essential nutrient Zn (49, 50). Despite this challenge, successful pathogens, such as *S. suis*, remain capable of causing severe disease. The success of *S. suis* and other invaders is mediated by an ability to compete with the host for Zn (37, 51). In bacteria, ABC transporters are used to transport metal ions. ABC transporters consist of a membrane-attached lipoprotein substrate binding protein, a membrane permease(s) and a ATPase proteins (20). Zn acquisition is mediated by ABC transporters identified by their lipoprotein components, the way that Zn-specific substrate binding protein achieve metal ion selection from the complex chemical environment of the host is unclear (52). Our

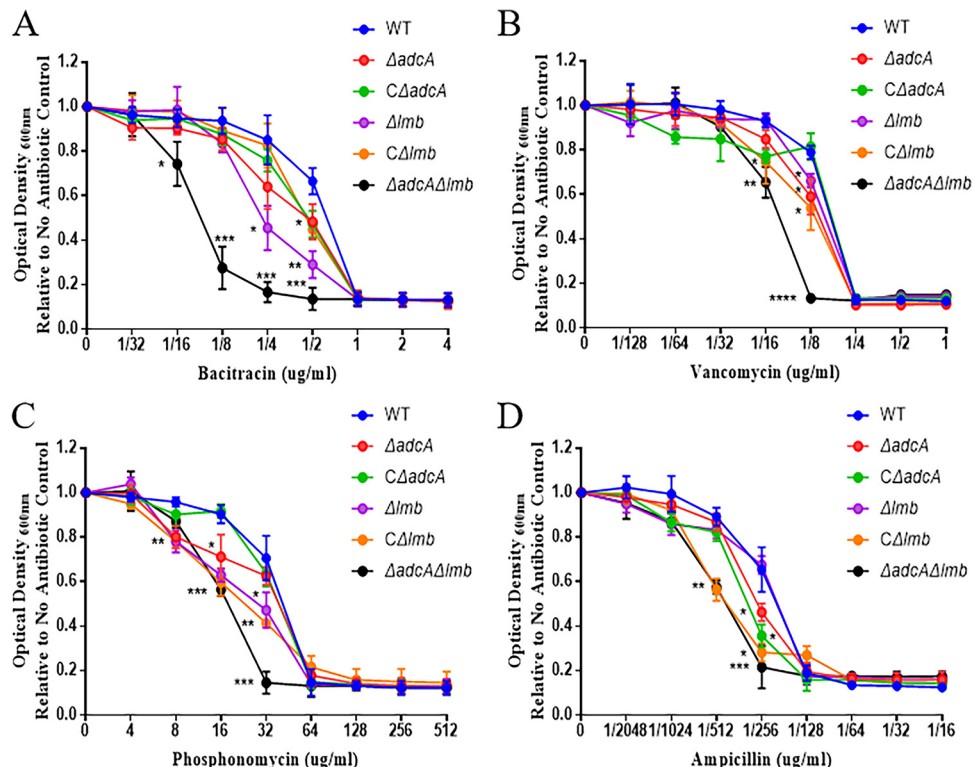

**FIG 7** Characterization of *S. suis* virulence traits at the cell surface interface. Final growth yields of WT, Δ*adcA*, CΔ*adcA*, Δ*lmb*, CΔ*lmb* and Δ*adcA*Δ*lmb* strains after 24 h of incubation in Mueller-Hinton agar (MH) supplemented with 2-fold increasing concentrations of (A) bacitracin, (B) vancomycin, (C) phosphonomycin, and (D) ampicillin. Data points represent the average of three biological replicates. Statistical analyses were performed using a two-tailed unpaired *t* test.

work reveals that *S. suis* possesses two distinct types of lipoprotein components, AdcA and Lmb, which have partly redundant role in Zn acquisition. AdcA is associated with direct recruitment of Zn. The *lmb* gene encodes a laminin, which has previously been associated with Zn acquisition. We found that the transcription levels of *adcA* and *lmb* in wild-type strain increased significantly in Zn-restricted medium (Fig. 2). The Δ*adcA*Δ*lmb* strain could not grow in Zn-restricted CDM medium (Fig. 3). The specific transport mechanism of AdcA and Lmb needs to be further elucidated. The regulation mechanism of Zn transport in *S. suis* needs to be further elucidated.

Many metal transporters have been reported to be associated with virulence in *S. suis*, Such as Mn uptake system TroABCD (53, 54), Mn efflux system MntE (55), Fe transporter FeoAB (56), cation-uptake regulators AdcR and Fur (57). In this manuscript, we have demonstrated that deletion of *adcA* or *lmb* partially attenuated virulence of *S. suis*, while deletion of both *adcA* and *lmb* had a profound effect on *S. suis* pathogenicity under low Zn conditions, as well as virulence in a murine model. In a murine model, when infected with lethal doses of *S. suis*, the surviving mice in WT group showed obvious clinical symptoms such as rough coat, depression, loss of appetite and weight loss. While all mice in the Δ*adcA*Δ*lmb* group survived and were in a good mental state (Fig. 8A). In addition, the colonization ability of Δ*adcA* and Δ*adcA*Δ*lmb* strains decreased significantly in all organs compared with WT strain, especially in lung (Fig. 8D to F). At last, we found Δ*adcA*Δ*lmb* was more easily cleared in the blood by the host compared with WT strain after infection (Fig. 8B and C). AdcA and Lmb has no influence on inflammatory cytokine production at an early stage of infection (Fig. 8I to K). Surprisingly, the Δ*adcA*Δ*lmb* group showed no obvious histopathological damage in lung, liver, spleen, brain compared to WT group, only a slight inflammatory cell exudation was observed (Fig. 9). This suggests that Zn plays an important role in the pathogenesis of *S. suis*, which

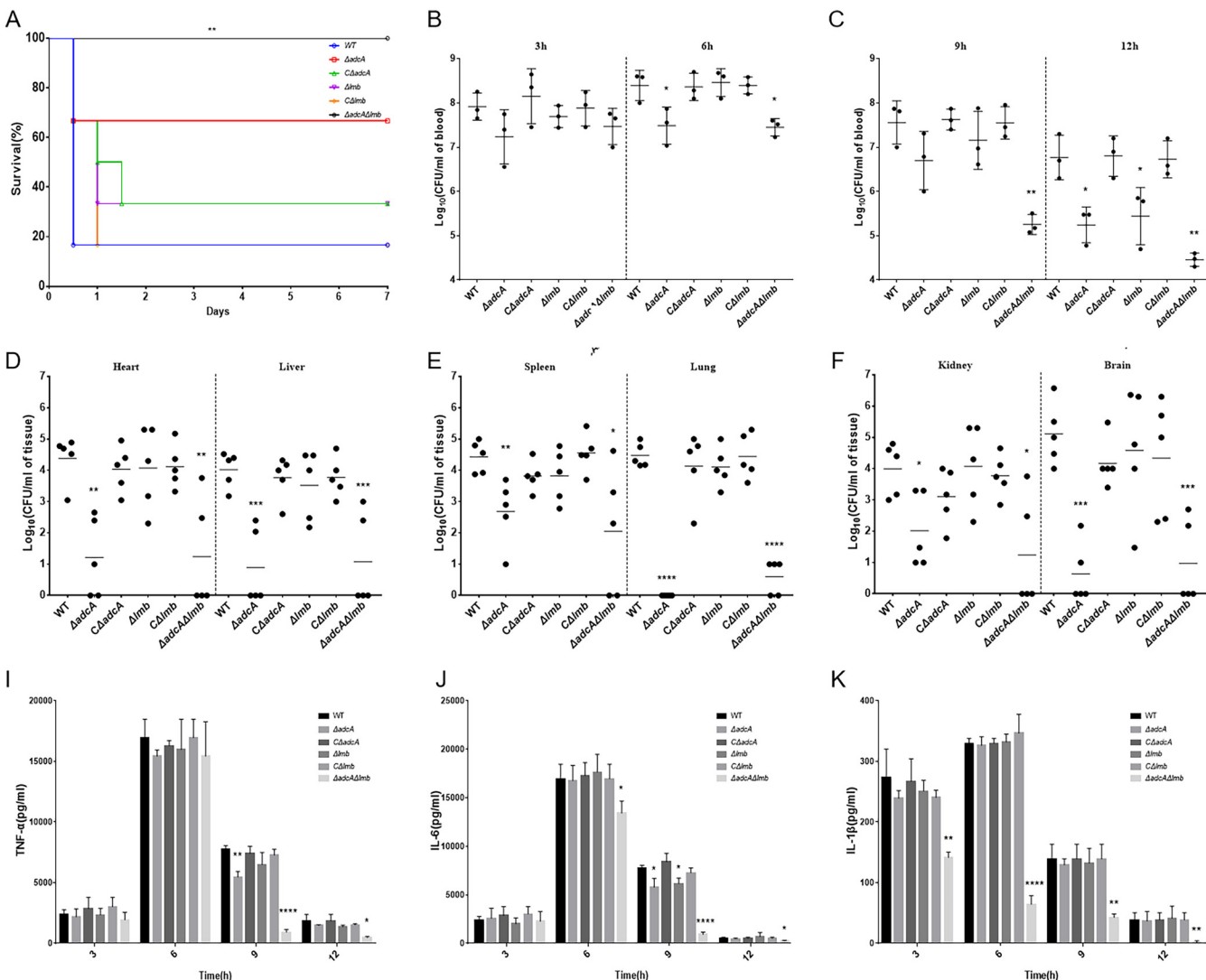

**FIG 8** Virulence of WT, Δ*adcA*, CΔ*adcA*, Δ*lmb*, CΔ*lmb*, and Δ*adcA*Δ*lmb* strains in a BALB/c mouse model. (A) Survival curves for mice infected with *S. suis* strains. Differences in survival rates between SC19 and the mutant strains were analyzed with a log-rank test (*P* < 0.01). (B and C) Bacterial burdens in the blood (CFU/mL of blood) at 3, 6 (B), 9, and 12 h (C). (D to F) Bacterial burdens in the heart and liver (D), spleen and lung (E), and kidney and brain (F) (CFU/g of tissue). (I and K) Concentrations of tumor necrosis factor α (TNF-α) (I), interleukin (IL)-6 (J), and IL-1β (K) in the serum at 3, 6, 9, 12 h post-infection; results are shown from three infected mice per group at each indicated time point. Statistical analyses were performed using a two-tailed unpaired *t* test.

may provide potential targets for potential inhibitors and therapeutic agents to control *S. suis* infections.

Biofilm formation by bacteria is one of the important reasons for chronic and persistent infections particularly difficult to cure (58). The biofilm may cause bacterial changes in drug resistance, acid resistance, and hunger resistance (13). Biofilm formation is likely contributing to the virulence and drug resistance in *S. suis*. At present, there are a variety of drug resistance mechanisms associated with the antibiotic resistance of *S. suis* biofilm (59). The biochemical factors mainly include quorum sensing system, extracellular polymeric substance (EPS) matrix, extracellular DNA, efflux pumps (9, 44, 60, 61). Besides, the low growth rate and metabolic adaptations are two key physiological factors that modulate drug resistance in biofilm (62, 63). Our study found that simultaneous deletion of *adcA* and *lmb* led to readily discernible morphological and biophysical alterations (Fig. 4). The Δ*adcA*Δ*lmb* strain shows reduced biofilm formation due to downregulation the expression of adhesion-related genes(Fig. 7). Furthermore, the Δ*adcA*Δ*lmb* strain shows more sensitive to bacitracin, vancomycin, phosphonomycin and ampicillin that

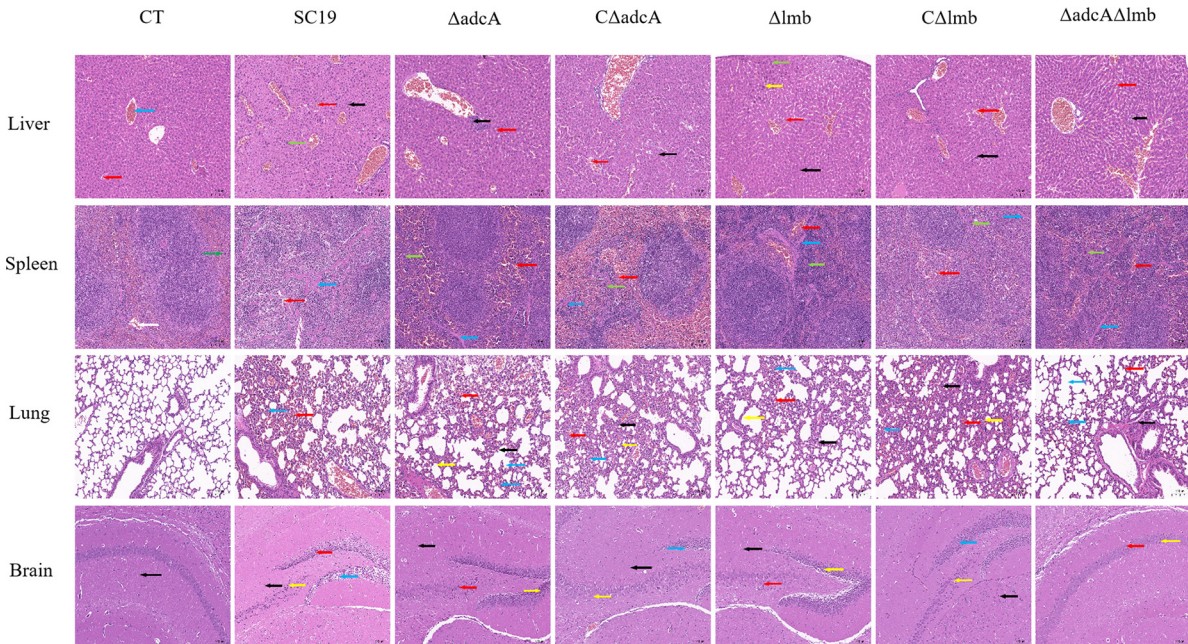

**FIG 9** Histopathology of *S. suis* infections caused by the WT, Δ*adcA*, CΔ*adcA*, Δ*lmb*, CΔ*lmb*, and Δ*adcA*Δ*lmb* strains. Control group mice were treated with the same dose of phosphate-buffered saline. Arrows indicate histopathological changes in the hematoxylin and eosin staining of lung, liver, spleen, and brain tissues. Liver tissues (lane 1): inflammatory cell infiltration (black arrows) and punctate necrosis (red arrows). A few liver cells are deeply stained (green arrow). Spleen tissues (lane 2): white pulp disturbed and indistinct from the red pulp (green arrows), bleeding in the splenic parenchyma (red arrows), splenic trabeculae (blue arrows), and spleen sinusoid (white arrow). Lung tissues (lane 3): telangiectasia, bleeding (red arrows), alveolar wall thickening (yellow arrows), alveolar atrophy (blue arrows), and inflammatory cell infiltration (black arrows). Brain tissues (lane 4): disappearance of pyramidal cells in the CA3 region of the hippocampus, neurofibrillary tangles in neurons (yellow arrow), neural cell degeneration in the dentate gyrus (DG) region (blue arrow), and heterotypic cell infiltration in the DG (red arrows), and gliocyte (black arrow). Magnification: ×100.

target the cell wall (Fig. 5). Therefore, both the substrate binding proteins AdcA and Lmb can be viewed as suitable targets for the development of antimicrobial therapies to treat or prevent *S. suis* infections.

In conclusion, we used bioinformatics and mutational analyses to identify and characterize the Zn uptake system in *S. suis*. Our results demonstrated that AdcA and Lmb are involved in Zn uptake in *S. suis*: they have overlapping roles in Zn acquisition and are essential for *S. suis* to overcome Zn limitation within the host. The Δ*adcA*Δ*lmb* strain shows reduced biofilm formation due to downregulated expression of adhesion-related genes. Surprisingly, Δ*adcA*Δ*lmb* showed more sensitivity to cell wall-targeting drugs by drug screening; the mechanism for this needs to be further studied in the future. In addition, the Zn uptake system is necessary for full virulence in *S. suis*. However, further studies are required to understand the contribution of the Zn efflux system to *S. suis* during infection. Collectively, our study revealed that Zn acquisition is essential for the pathogenesis of *S. suis* and Zn uptake systems may be targets for the development of new antimicrobials.

## MATERIALS AND METHODS

**Bacterial strains, plasmids, primers and growth condition.** Bacterial strains and plasmids used in this study are listed in Table 1. Primers are listed in Table 2. The *S. suis* strain was cultured at 37°C on brain heart infusion (BHI, Oxoid), tryptic soy broth (TSB; Becton Dickinson [BD]) or tryptic soy agar (TSA; BD) containing 10% (vol/vol) newborn bovine serum. The blood of healthy pigs came from a pig farm in Hubei Province. The preparation of the CDM was carried out as previously described (64). The phosphate-buffered saline (PBS) was treated with 0.25 mM EDTA. The *Escherichia coli* competent cell lines, DH5α and BL21, were grown in Luria-Bertani broth (LB) or on LB agar at 37°C. When required, spectinomycin (Sigma) was added at the following concentrations: 50 mg/mL for *E. coli* and 100 mg/mL for *S. suis*.

**Bioinformatic analysis.** BLASTP searches were conducted to identify the homologues of AdcA and Lmb in the *S. suis* genome. Multiple sequence alignments were performed using Clustal Omega

**TABLE 1** Bacterial strains, plasmids used in this study

| Strain or plasmid | Relevant characteristics[a] | Source or reference |
|---|---|---|
| **Strains** | | |
| SC19 | Virulent *S. suis* strain isolated from brain of dead pig | Storage in laboratory |
| Δ*adcA* | *adcA* deletion mutant of strain SC19 | This study |
| CΔ*adcA* | Complemented strain of Δ*adcA* mutant; Spc$^r$ | This study |
| Δ*lmb* | *lmb* deletion mutant of strain SC19 | This study |
| CΔ*lmb* | Complemented strain of Δ*lmb* mutant; Spc$^r$ | This study |
| Δ*adcA*Δ*lmb* | *adcA* and *lmb* deletion mutant of strain SC19 | This study |
| DH5$\alpha$ | Cloning host for recombinant vector | Tsingke |
| BL21 | Expression host for protein | Tsingke |
| | | |
| **Plasmids** | | |
| pSET4s | Thermosensitive suicide vector; Spc$^r$ | 67 |
| pSET4s-Δ*adcA* | Knockout vector for *adcA* deletion; Spc$^r$ | This study |
| pSET4s-Δ*lmb* | Knockout vector for *lmb* deletion; Spc$^r$ | This study |
| pSET2 | *E. coli-S. suis* shuttle vector; Spc$^r$ | 65 |
| pSET2-*adcA* | pSET2 containing *adcA* and its promoter; Spc$^r$ | This study |
| pSET2-*lmb* | pSET2 containing *lmb* and its promoter; Spc$^r$ | This study |
| pET-28a | Expression vector for AdcR; Kan$^r$ | Laboratory storage |
| pET-30a | Expression vector for calprotectin; Kan$^r$ | Laboratory storage |

[a]Spc$^r$, spectinomycin resistant; Kan$^r$, kanamycin resistant.

(https://www.ebi.ac.uk/Tools/msa/clustalo/). Promoters were predicted using BPROM (http://linux1 .softberry.com/berry.phtml).

**Growth curve analyses.** The WT, Δ*adcA*, Δ*lmb*, and Δ*adcA*Δ*lmb* strains were grown in TSB medium to the early stationary phase (OD$_{600}$ [optical density at 600 nm] of 1.2), then washed three times with PBS (containing 0.25 mM EDTA) and diluted in fresh medium supplemented with various concentrations of the specified metals. For the Zn sensitivity assay, overnight cultures of the WT, Δ*adcA*, Δ*lmb*, and Δ*adcA*Δ*lmb* strains were diluted 1:100 in CDM medium supplemented with different concentrations of ZnSO$_4$ (0.0001, 0.001, and 0.01 mM) and incubated at 37°C under static conditions. Aliquots were taken from the cultures to measure the OD$_{600}$ every hour. For the metal sensitivity assay, overnight cultures of the WT, Δ*adcA*, Δ*lmb*, and Δ*adcA*Δ*lmb* strains were diluted 1:100 in CDM medium supplemented with different metal ions and incubated at 37°C under static conditions. Aliquots were taken from the cultures to measure the OD$_{600}$ every hour. The metal ions tested were copper (II) (CuSO$_4$·5 H$_2$O), iron (II) (FeSO$_4$·7 H$_2$O), manganese (II) (MnSO$_4$·H$_2$O), nickel (II) (NiSO$_4$·6 H$_2$O), zinc (II) (ZnSO$_4$·7 H$_2$O) and cobalt (II) [Co (NO$_3$)$_2$·6 H$_2$O].

**RNA extraction, RT-PCR, and reverse transcription-quantitative PCR.** The total RNA of bacteria was extracted using a Bacteria Total RNA isolation kit (Sangon Biotech, China) and the HiScript II Q RT SuperMix for reverse transcription-quantitative PCR (RT-qPCR) (+gDNA wiper) (Vazyme, China) was used to synthesize cDNA. A one-step reaction in a ViiTM7 real-time PCR system was used to perform the quantitative PCR. AceQ qPCR SYBR Green Master Mix (Vazyme, China) was used to measure the mRNA levels according to the manufacturer's instructions.

**Construction of the deletion mutant and complementation strain.** To obtain a markerless deletion mutant, we used the R1/R2 and L1/L2 primers to separately amplify the upstream and downstream regions of the target genes by PCR. The overlapping PCR products were directly cloned to a pSET4s vector following digestion with the corresponding restriction enzymes. The recombinant plasmid was transformed into the *S. suis* strain by electroporation. After two steps of allelic exchange, spectinomycin-sensitive clones were selected, and the mutant was identified by PCR using two pairs of specific primers listed in Table 2. The mutant was further verified by RT-PCR and DNA sequencing analysis. The complementation strains were generated by the *E. coli-S. suis* shuttle vector pSET2 (65), as previously described (66). A DNA fragment containing the gene and its predicted promoter was amplified from the *S. suis* genome and cloned into pSET2 to generate plasmids pSET2: *adcA* and pSET2: *lmb*. These plasmids were electroporated into the corresponding mutant strains and the complementation strains were selected with spectinomycin. The complementation strains were further confirmed by PCR, RT-PCR, and DNA sequencing analysis.

**Scanning electron microscopy analysis of bacterial cell morphology.** WT and Δ*adcA*Δ*lmb* strains were grown in CDM with 0.01 mM ZnSO$_4$ overnight, washed three times in zinc-restricted CDM, and inoculated at OD$_{600}$ = 0.2 in 15 mL Zn-deprived (without Zn) or Zn-containing CDM (with 0.01 mM Zn) for 5.5 h. The bacteria were harvested by centrifugation and cells were washed three times with PBS containing 0.25 mM EDTA. Next, bacteria were fixed with 2.5% glutaraldehyde at 4°C overnight. The samples were then treated with 1% osmium tetroxide for 2 h at room temperature and dehydrated in a serial dilution of ethanol. The dehydrated cells were coated with a 10-nm-thick gold layer for 30 s and observed by a SU8100 scanning electron microscope (Hitachi, Japan).

**Adherence and invasion assay.** Cells were seeded in 24-well plates. The WT and deletion mutant strains were added to hBMEC and hep-2 cell monolayers, respectively, at an MOI of 100:1 at 37°C, 5% CO$_2$. After 2 h, the hBMEC monolayers were washed three times with PBS and lysed with 0.025% Triton X-100 on ice for 10 min. The number of bacteria adhering to hBMEC was calculated. The hep-2 cell

**TABLE 2** Primers used in this study

| Primer | Sequence (5′→3′)[a] | Size (bp) | Target gene |
|---|---|---|---|
| L1-*adcA* | AAAA**CTGCAG**GATCCTGATGAGCTGAATAAGTATG | 1,023 | The left arm of *adcA* |
| L2-*adcA* | CCAACAGGGTAGTATGTCGGTTGCCACAAGCACCCAAAAG | | |
| R1-*adcA* | CTTTTGGGTGCTTGTGGCAACCGACATACTACCCTGTTGG | 1,139 | The right arm of *adcA* |
| R2-*adcA* | CCG**GAATTC**CAAATTTCACTTGCACAAGCGCAAC | | |
| In-*adcA*-F | CTCTTGAAAGTTTGACGGATGA | 675 | An internal region of *adcA* |
| In-*adcA*-R | GAGCCAGCATCTCCTGACCAAT | | |
| Out-*adcA*-F | ACATGCATGCTTTGTAGATGGGTTACCTGTGCGAG | 1,962 | A fragment containing *adcA* |
| Out-*adcA*-R | ATTGTCTGCTCCACACATTCACCTC | | |
| C-*adcA*-F | ACAT**GCATGC**TTTGTAGATGGGTTACCTGTGCGAG | 1,881 | *adcA* and its promoter |
| C-*adcA*-R | CG**GAATTC**ACACCTTGCCCAGTCTTCTCATTAT | | |
| L1-*lmb* | AA**CTGCAG**TTACAAAGAAACTCTCTGAAAAACC | 1,035 | The left arm of *lmb* |
| L2-*lmb* | AGTGTAAGAGAAGGCTGTGTGTTGATTTCTTTAACATAACTTCCTCCTTT | | |
| R1-*lmb* | AAAGGAGGAAGTTATGTTAAAGAAAAATGAAGAAAAAGCAGTTGTTGGC | 717 | The right arm of *lmb* |
| R2-*lmb* | ACAT**GCATGC**ATTCCAGTAATCTTGAGCAGCCTTC | | |
| In-*lmb*-F | ATGTTAAAGAAAGTGATAAGAGGCT | 390 | An internal region of *lmb* |
| In-*lmb*-R | AGCTTCCATGTCTTCTAACCCT | | |
| Out-*lmb*-F | AACTGCAGGCAAGATTTATTTATTGGACCTATG | 1,359 | A fragment containing *lmb* |
| Out-*lmb*-R | ATAGGACACACGATTTTTTTGCTCT | | |
| C-*lmb*-F | AA**CTGCAG**GCAAGATTTATTTATTGGACCTATG | 1,223 | *lmb* and its promoter |
| C-*lmb*-R | CGGAATTCGGCCTTTTATTTTAACTCTTGAGCT | | |
| Q16S-F | ACTTGAGTGCAGAAGGGGAGAG | 107 | An internal region of 16S rRNA |
| Q16S-R | GCGTCAGTTACAGACCAGAGAGC | | |
| Q*adcA*-F | TTTGGCTTTGGACTATGGTTTG | 153 | An internal region of *adcA* |
| Q*adcA*-R | ACAGATTTTGACGCATTTTCTT | | |
| Q*lmb*-F | CCAGTTTTGGTTGGTCAGGAAG | 143 | An internal region of *lmb* |
| Q*lmb*-R | AAGATTGGGCTATACTTGTCTGC | | |
| Q*ccpa*-F | CGGTGTCAGTGATATGGG | 96 | An internal region of *ccpa* |
| Q*ccpa*-R | GTCAGGTTTGGACGGGTA | | |
| Q*fbps*-F | AACCATCTTGCCAGGCTCCAC | 169 | An internal region of *fbps* |
| Q*fbps*-R | CAGTTCAGAAGCCGTATCCCGAC | | |
| Q*gapdh*-F | CTTGGTAATCCCAGAATTGAACGG | 134 | An internal region of *gapdh* |
| Q*gapdh*-R | TCATAGCAGCGTTTACTTCTTCAGC | | |
| Q*gdh*-F | CACCTTTACCACCGCCGATTG | 175 | An internal region of *gdh* |
| Q*gdh*-R | GGAAATGTTCAAGTCAACCGTGG | | |
| Q*gor*-F | GTTCACGCGCATCCTACG | 171 | An internal region of *gor* |
| Q*gor*-R | TACCAGGAATAGCAGGGAC | | |
| Q*perR*-F | TTGAACACGTCATCCAACAT | 200 | An internal region of *perR* |
| Q*perR*-R | GTAGTTAGGTATTAGATCTTG | | |
| Q*srtA*-F | AGGCGAAACAATTTCCACAC | 201 | An internal region of *srtA* |
| Q*srtA*-R | GGAGCTGGTACCATGAAGGA | | |
| Q*gdpP*-F | CTTCTGCGATTGTCTGGTCA | 164 | An internal region of *gdpP* |
| Q*gdpP*-R | AATTGAGGCGGTATTCGTTG | | |

[a]Bold, underlined nucleotides represent restriction sites.

monolayers were washed three times with PBS. Next, the cells were exposed to medium containing gentamicin (100 $\mu$g/mL) and penicillin-G (0.5 $\mu$g/mL) for 1 h to kill extracellular bacteria; the cells were then washed again and the number of bacteria invading hep-2 cells was calculated.

**Biofilm assay.** Bacteria were grown in TSB at 37°C to the stationary growth phase and then diluted 1:100 in TSB. The biofilm assay was performed using 24-well polystyrene plates. The microplate was incubated at 37°C for 24 h. The medium was replaced after 12 h. After incubation, the spent medium was discarded, and biofilms were washed three times with PBS and stained with 1% crystal violet for 30 min. A 33% acetic acid solution was used to dissolve the precipitated crystal violet-stained biomass, and absorbance was determined at $OD_{590}$ using a Victor Nivo multifunctional enzyme reader (Perkin Elmer, USA). All assays were done in triplicate, and three independent experiments were performed.

**MIC determinations.** The minimum inhibitory concentration (MIC) assays were conducted according to Clinical and Laboratory Standards Institute (CLSI) guidelines. Overnight bacterial cultures were diluted 1:100 with fresh TSB and grown until the $OD_{600}$ reached 0.6. Then, the cells were washed three times with PBS, the bacterial concentration was adjusted to $1 \times 10^6$ CFU/mL, and the diluted cultures were inoculated at a ratio of 1:1 into Mueller-Hinton agar (MH) containing different concentrations of antibiotics at 37°C for 24 h (ampicillin, bacitracin, daptomycin, and vancomycin). The absorbance at $OD_{600}$ was measured using a Victor Nivo multifunctional enzyme reader (Perkin Elmer).

**Pig blood killing assay.** Overnight bacterial cultures were diluted 1:100 with fresh TSB and grown until the $OD_{600}$ reached 0.6. Next, the cells were washed there times with PBS, the bacterial concentration was adjusted to $1 \times 10^7$ CFU/mL, and 100 $\mu$L of bacterial suspension was mixed with 900 $\mu$L of

healthy pig whole blood and incubated at 37°C for 3 h. Next, 100 $\mu$L of bacterial solution was removed at 0, 1, 2, and 3 h, respectively, for plate counting. The growth factor was defined as the ratio of the number of viable bacteria in the sample after 1, 2, and 3 h of incubation, respectively, to the number of viable bacteria after 0 h incubation.

**Animal experiments.** All animal studies were approved by the Laboratory Animal Monitoring Committee of Huazhong Agricultural University and conformed to the recommendations in the Guide for the Care and Use of Laboratory Animals of Hubei Province, China. A total of 60 female BALB/c mice (5 weeks old) were randomly separated into six groups. Six groups of BALB/c mice (10 mice per group) were inoculated by an intraperitoneal injection of $5 \times 10^8$ CFU of the WT, Δ*adcA*, CΔ*adcA*, Δ*lmb*, CΔ*lmb*, and Δ*adcA*Δ*lmb* strains. The mortality rates and mental state of mice for each group were recorded each day.

To evaluate the colonization ability of wild-type and mutant strains in the mice, 36 BALB/c mice (5 weeks old) were randomly separated into six groups. Six groups of BALB/c mice (6 mice per group) were inoculated by an intraperitoneal injection of $2 \times 10^8$ CFU of the WT, Δ*adcA*, CΔ*adcA*, Δ*lmb*, CΔ*lmb*, and Δ*adcA*Δ*lmb* strains. After 48 h, blood was removed from each mouse via the orbital vein, and the heart, liver, spleen, lung, kidney, and brain were aseptically removed for bacterial counts and histopathological section observation. Homogenized tissues were plated on TSA plates containing 10% (vol/vol) newborn bovine serum to determine bacterial loads.

To evaluate the production of inflammatory cytokines of wild-type and deletion-mutant strains in the mice, 48 BALB/c mice (5 weeks old) were randomly separated into four groups. For each group, $2 \times 10^8$ CFU bacteria per mouse were injected via the abdominal cavity. Every 3 h, 3 mice were randomly removed from each group for orbital vein blood collection and then euthanized. The blood of each mouse was used for viable counts and detection of inflammatory cytokines. Blood was plated on TSA plates containing 10% (vol/vol) newborn bovine serum to determine bacterial loads. The level of serum inflammatory cytokines were detected by an enzyme-linked immunosorbent assay (ELISA) kit (4A Biotech).

**Statistical analysis.** GraphPad Prism 7 software was used to analyze the data. A Student's *t* test analysis of variance was used to analyze the results. For all tests, $P < 0.05$ was considered the threshold for significance.

## ACKNOWLEDGMENTS

We thank Tsutomu Sekizaki (National Institute of Animal Health, Japan) for supplying the plasmids pSET4s and pSET2.

This research was funded by the National Key Research and Development Program of China (2021YFD1800400; 2021YFD1800401), the Natural Science Foundation of China (NSFC; grant no. 31672560 and 32273039), the Technical Innovation Project of Hubei Province (2021ABA005), the Hubei Province Natural Science Foundation for Innovative Research Groups (2021CFA019), the Science and Technology Project of Hubei Province (2022BBA0055), and the Hubei Province Innovation Center of Agricultural Sciences and Technology (2019-620-000-001-017). The funders had no role in the study design, data collection and analysis, decision to publish, or preparation of the manuscript.

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
