## [Reviewer comments · Microbiology Spectrum]

Microbiology Spectrum

The *adcA* and *Imb* gene play an important role in drug resistance and full virulence of *Streptococcus suis*

Weicheng Bei, Fangyan Yuan, mingzheng peng, Yuanyuan Xu, Yongxiang Tian, Danna Zhou, Keli Yang, Zewen Liu, Wei Liu, Ting Gao, and Chang Li

Corresponding Author(s): Weicheng Bei, Huazhong Agricultural University

Review Timeline:

Submission Date:	October 25, 2022
Editorial Decision:	November 29, 2022
Revision Received:	February 9, 2023
Accepted:	February 25, 2023

Editor: Ahmed Babiker

Reviewer(s): The reviewers have opted to remain anonymous.

Transaction Report:

DOI: <https://doi.org/10.1128/spectrum.04337-22>

November 29, 2022

Prof. Weicheng Bei
Huazhong Agricultural University
Huazhong Agricultural University
Wuhan
China

Re: Spectrum04337-22 (The *adcA* and *lmb* gene play an important role in drug resistance and full virulence of *Streptococcus suis*)

Dear Prof. Weicheng Bei:

Thank you for submitting your manuscript to Microbiology Spectrum. At this time we cannot accept the paper. The reviewers have raised major concerns with which I agree. We can consider a revised version of the manuscript which adequately addresses the reviewers concerns. When submitting the revised version of your paper, please provide (1) point-by-point responses to the issues raised by the reviewers as file type "Response to Reviewers," not in your cover letter, and (2) a PDF file that indicates the changes from the original submission (by highlighting or underlining the changes) as file type "Marked Up Manuscript - For Review Only". Please use this link to submit your revised manuscript - we strongly recommend that you submit your paper within the next 60 days or reach out to me. Detailed instructions on submitting your revised paper are below.

Link Not Available

Sincerely,

Ahmed Babiker

Journals Department
Reviewer comments:

Reviewer #1 (Comments for the Author):

This manuscript describes the role of AdcA and Lmb, two zinc-binding proteins in *Streptococcus suis*. Although this study is rather complete in term of methodologies, a large part of the results were already published (see "The AdcR-regulated AdcA and AdcAll contribute additively to zinc acquisition and virulence in *Streptococcus suis*", Zheng et al. 2022). I also have concerns concerning some data, in particular with the results on biofilms (see below). Finally the english is sometimes of poor quality, there is a lack of references and a lot of mistakes which requires a more careful re-reading of the manuscript. You can find below some suggestions that should help the authors to improve their manuscript.

Line 19, 33, 70, 87, 90, 304, 338, etc...: A bacteria name must be in italic with a capital letter for the genus name and with a

space between the genus name and the specie.

line 22: *adcA* or *adcACB* mutant strain ?

line 23: Zinc-limited , in lowercase characters

line 42 and 78: *AdcA* and *Lmb* are zinc-binding proteins and not transporters as the complete transporter also comprise the integral membrane protein, *AdcB*, a nucleotide binding domain, *AdcC* and histidine triad (HT) proteins (Plumptre et al. 2014; Moulin et al. 2019)

line 44, 128, 429: sensitivity or susceptibility and not sensitive

line 77: Plumptre et al. 2014 and Moulin et al. 2019 also highlighted two distinct routes of zinc acquisition in *Streptococci* mediated by either *AdcA* alone, or via *Lmb/AdcAll* in concert with the *Sht*-family proteins. These modes of zinc acquisition have been attributed to the distinct structural features of *AdcA* and *AdcAll*. This should be mentioned here and used to describe more precisely *AdcA* and *Lmb* of *S. suis*.

line 121: the sentence has no sense

Line 271-276 and figure 2: Should be removed from the results section and place in material and methods.

Line 275: This is the first figure, so that it should be numbered as figure 1, etc...

Line 279: a part of the sentence is missing

Fig 5A and B: Both complementation strains do not fully complement. How do you explain this? Moreover, in TSB medium, in which biofilm experiments were conducted, both *lmb* and *adcA* gene are repressed because of the zinc. This should be at least discussed.

Fig. 5C: there is no mention of the growth conditions for the q-RT PCR experiments: in which medium ? At which phase of growth?

line 328-330: to my knowledge none of the genes that were tested here is directly linked to adhesion. For example *CcpA* and *PerR* are transcriptional regulators controlling respectively carbon metabolism and metal homeostasis and can't be named "adhesion genes". Did they test others real "adhesion gene", like gene encoding *pili* ?

In Fig.5, most of the differences are below a 2-fold change. I am not convinced by the biological relevance of these differences. Moreover the title "*AdcA* and *Lmb* contribute to biofilm formation in *S. suis* by downregulating the expression of adhesion genes" is not supported by the data. There is no evidence that the difference in *perR* and *gdh* expression has any connection with the difference in biofilm formation. Additional experiments are needed to prove it.

Fig. 8: Unfortunately, similar survival curves and bacterial counts in organs have been published in June 2022 by Zheng et al..

Line 349: remove "was" in the sentence

Line 388 "*AdcA* is associated with direct recruitment of

Zn " : in which bacteria ? Reference ?

Line 389 "*The lmb* gene encodes a laminin, which has previously been associated with Zn acquisition". *Lmb* certainly not encodes a laminin ! It is a laminin-binding protein. Reference ?

Line 394 "*The regulation mechanism of Zn transport in S. suis* needs to be further elucidated": in numerous *Streptococcal* species, the *Adc/lmb* transporter is regulated by *AdcR* (Shafeeq et al. 2011 ; Sanson et al. 2015 ; Moulin et al. 2016). This is certainly also the case in *S. suis* and it should be discussed here.

Line 417: "nutrient availability" rather than "hunger resistance"

Line 428: The authors did not provide any reference linking the tested genes with adhesion. Moreover their data do not show that the reduced biofilm formation is due to the downregulation of these genes.

Reviewer #2 (Comments for the Author):

The manuscript by Mingzheng Peng et al. provides a systematic analysis of the phenotype displayed by knock-out mutants of the genes *adcA* and *lmb*, coding for the two Zn transporters in *Streptococcus suis*.

The manuscript presents a useful study on the influence of zinc homeostasis in cell adhesion, invasion and biofilm formation by *S. suis* as well as on its tolerance towards antibiotics. However, a more detailed genetic characterization of the strain used should be reported and discussed as the manuscript suffers from the lack of a thorough presentation and analysis of the data. For instance, there is no mention of the source of the strain used in the study or reference to its genome sequence, which has been present in the NCBI database since 2017.

Furthermore, a clear comprehension of the work is hindered by the poor quality of the English language usage and punctuation. In addition, the inaccurate use of technical terminology renders the manuscript more difficult to follow.

For example:

line 273 "complementary strains" instead of "complementation strains"

line 279 strain in TSB. To explore there should be a comma instead of a full stop

I strongly suggest that the manuscript should be checked by an English-speaking colleague, the clarity of the topics presented would benefit significantly.

In conclusion, the manuscript offers an extensive characterisation of the effects of loss of zinc homeostasis in *S. suis*, but the authors must capitalise better the considerable experimental work by presenting their study in a more careful manner and provide a more complete discussion of their results compared to the many relevant reports available in the literature.

Staff Comments:

Preparing Revision Guidelines

Please return the manuscript within 60 days; if you cannot complete the modification within this time period, please contact me. If you do not wish to modify the manuscript and prefer to submit it to another journal, please notify me of your decision immediately so that the manuscript may be formally withdrawn from consideration by Microbiology Spectrum.

The manuscript by Mingzheng Peng et al. provides a systematic analysis of the phenotype displayed by knock-out mutants of the genes *adcA* and *lmb*, coding for the two Zn transporters in *Streptococcus suis*.

The manuscript presents a useful study on the influence of zinc homeostasis in cell adhesion, invasion and biofilm formation by *S. suis* as well as on its tolerance towards antibiotics. However, a more detailed genetic characterization of the strain used should be reported and discussed as the manuscript suffers from the lack of a thorough presentation and analysis of the data. For instance, there is no mention of the source of the strain used in the study or reference to its genome sequence, which has been present in the NCBI database since 2017.

Furthermore, a clear comprehension of the work is hindered by the poor quality of the English language usage and punctuation. In addition, the inaccurate use of technical terminology renders the manuscript more difficult to follow.

For example:

line 273	“complementary strains”	instead of “complementation strains”
line 279	strain in TSB. To explore	there should be a comma instead of a full stop

I strongly suggest that the manuscript should be checked by an English-speaking colleague, the clarity of the topics presented would benefit significantly.

In conclusion, the manuscript offers an extensive characterisation of the effects of loss of zinc homeostasis in *S. suis*, but the authors must capitalise better the considerable experimental work by presenting their study in a more careful manner and provide a more complete discussion of their results compared to the many relevant reports available in the literature.

Response to Reviewers

We would like to express our sincere thanks to the reviewers for their constructive comments and suggestions, which help us in depth to improve the quality of the manuscript. We have tried our best to revise the manuscript according to the comments. The following response is point-by-point towards the reviewers' comments

Reviewer comments:

Reviewer #1 (Comments for the Author):

This manuscript describes the role of AdcA and Lmb, two zinc-binding proteins in *Streptococcus suis*. Although this study is rather complete in term of methodologies, a large part of the results were already published (see "The AdcR-regulated AdcA and AdcAII contribute additively to zinc acquisition and virulence in *Streptococcus suis*", Zheng et al. 2022). I also have concerns concerning some data, in particular with the results on biofilms (see below). Finally the english is sometimes of poor quality, there is a lack of references and a lot of mistakes which requires a more careful re-reading of the manuscript. You can find below some suggestions that should help the authors to improve their manuscript.

RE: Thank you for your nice suggestion.

(1) Line 19, 33, 70, 87, 90, 304, 338, etc...: A bacteria name must be in italic with a capital letter for the genus name and with a space between the genus name and the specie.

RE: Thank you very much for your reminder. We have fixed these corresponding errors in the revised manuscript. Please refer to line 19, 32, 70, 88, 90, 93, 96, 99, 310 and 344 in the revised manuscript.

(2) line 22: *adcA* or *adcACB* mutant strain ?

RE: Thank very much for your advice. We constructed an *adcA* deficient strain in this study, we have replaced *adcACB* with *adcA* in the revised manuscript. Please refer to lines 22 in the revised manuscript.

(3) line 23: Zinc-limited , in lowercase characters

RE: Thank you for your nice suggestion. We have replaced Zinc-limited with zinc-limited in the revised manuscript. Please refer to lines 24 in the revised manuscript.

(4) line 42 and 78: AdcA and Lmb are zinc-binding proteins and not transporters as the complete transporter also comprise the integral membrane protein, AdcB, a nucleotide binding domain, AdcC and histidine triad (HT) proteins (Plumptre et al. 2014; Moulin et al. 2019)

RE: Thank you for your nice suggestion. We have replaced zn transporter with zinc-binding proteins in the revised manuscript. In *Streptococcus pneumoniae*, *Streptococcus pyogenes*, *Streptococcus agalactiae*, and *Streptococcus mutans*, zinc acquisition has been reported to be mediated by the AdcR regulon, that includes genes encoding: an ABC transporter AdcABC (comprising AdcA, a Zn-binding lipoprotein; AdcB, a permease; and AdcC, an ATP-binding protein), AdcAII (another Zn-binding lipoprotein), and several other proteins (Bayle et al., 2011; Makthal et al., 2020, 2017; Moulin et al., 2016; Pan et al., 2021; Plumptre et al., 2014; Sanson et al., 2015; Shafeeq et al., 2011). Most streptococci species encode two Zn-binding lipoprotein. Please refer to lines 41, 88 and 409 in the revised manuscript.

(5) line 44, 128, 429: sensitivity or susceptibility and not sensitive

RE: Thank you for pointing out our negligence. We have replaced sensitive with sensitivity or susceptibility. Please refer to lines 44, 136 and 464 in the revised manuscript.

(6) line 77: Plumptre et al. 2014 and Moulin et al. 2019 also highlighted two distinct routes of zinc acquisition in *Streptococci* mediated by either AdcA alone, or via Lmb/AdcAII in concert with the Sht-family proteins. These modes of zinc acquisition have been attributed to the distinct structural features of AdcA and AdcAII. This should be mentioned here and used to describe more precisely AdcA and Lmb of *S. suis*.

RE: Thank you for your nice advice. We have rewritten the statement in the revised manuscript. It has been reported zinc acquisition in *Streptococci* is mediated by either AdcA alone, or via Lmb/AdcAII in concert with the Sht-family proteins (Plumptre et al. 2014) (Moulin et al. 2019). In *streptococcus pneumoniae*, the AdcAN domain is necessary and sufficient for Zn²⁺ acquisition, with the AdcAC domain and the his-rich loop aiding in Zn²⁺ recruitment during growth under Zn²⁺-restricted conditions. Zn²⁺ uptake in the Adc permease is regulated by the AdcAN domain of AdcA (Zhenyao Luo et al., 2020). Intriguingly, this feature is absent from AdcAII, AdcAII is reliant upon the polyhistidine triad proteins for zinc in vitro and in vivo. This suggests that the two zinc-binding SBPs of *S. pneumoniae* employ different modalities in zinc recruitment (Plumptre et al. 2014). We added this content to the introduction section of the article. Please refer to lines 78-87 in the revised manuscript.

(7) line 121: the sentence has no sense

RE: Thank you for your advice. We have deleted this improper sentence "This study aimed to identify the role of Zn acquisition systems in *S. suis*." in the revised manuscript. Please refer to lines 130-131 in the revised manuscript.

(8) Line 271-276 and figure 2: Should be removed from the results section and place in material and methods.

RE: Thank you for your nice suggestion. We have changed this improper title into "Verification of the deletion mutant and complementation strain in the revised manuscript.". Please refer to line 280 in the revised manuscript. This is part of our results, we have described the relevant experimental procedures in material and methods. Please refer to lines 182-197 in the revised manuscript.

(9) Line 275: This is the first figure, so that it should be numbered as figure 1, etc...

RE: Thank you for your nice suggestion. We numbered all the pictures in the revised manuscript. Figure number have been updated. The figure legends have been carefully edited in the revised manuscript. They were put in a file named figure.

(10) Line 279: a part of the sentence is missing.

RE: Thank you for your nice suggestion. We have replaced this sentence with "The transcription levels of *adcA* and *lmb* in CDM medium with or without Zn were tested to explore the role of AdcA and Lmb of *S. suis* in Zn-restricted environments." in the revised manuscript. Please refer to lines 286-287 in the revised manuscript.

(11) Fig 5A and B: Both complementation strains do not fully complement. How do you explain this?

RE: Thank you for your nice suggestion. We repeated the biofilm experiment several times with the same results. As we complement the gene by transferring plasmid. We found that the expression level of related genes in the complement strain was lower than that in the wild strain by RT-PCR, which might explain that both complementation strains do not fully complement.

(12) Moreover, in TSB medium, in which biofilm experiments were conducted, both *lmb* and *adcA* gene are repressed because of the zinc. This should be at least discussed.

RE: Thank you for your nice suggestion. It has been reported that zinc ions are important for biofilm formation (Lam et al., 2022) (Li et al., 2022). In the previous experiment, we found that there was no significant difference in the growth of wild strains and mutant strains in TSB medium. Although the medium contains a certain concentration of zinc ions, the bacterial intake of zinc ions from the environment is reduced due to the absence of zinc uptake proteins AdcA and Lmb, thus affecting biofilm formation. We added this content to the discussion section of the article. Please refer to lines 443-449 in the revised manuscript.

(13) Fig. 5C: there is no mention of the growth conditions for the q-RT PCR experiments: in which medium ? At which phase of growth?

RE: Thank you for your nice suggestion. The WT or $\Delta adcA\Delta lmb$ strains were grown in TSBS medium overnight, then washed three times with PBS, overnight bacterial culture was diluted 1:100 with fresh TSB and grown until the OD600 reached 0.6. Then the total RNA of bacteria was extracted. We have added this condition in the revised manuscript. Please refer to lines 3-4 in the figure 5.

(14) line 328-330: to my knowledge none of the genes that were tested here is directly linked to adhesion. For example CcpA and PerR are transcriptional regulators controlling respectively carbon metabolism and metal homeostasis and can't be named "adhesion genes". Did they test others real "adhesion gene", like gene encoding pili ?

RE: Thank you for your Correction. We have changed this part into "AdcA and Lmb contribute to biofilm formation of *S. suis* by influencing bacterial virulence" in the revised manuscript. It has been reported that biofilm formation is related to the bacterial virulence (Lam et al., 2022) (Wang et al., 2019b). We repeated the q-RT PCR experiment of eight virulence-related genes (*gapdH*, *srtA*, *gdpP*, *perR*, *gdh*, *gor*, *ccp* and *fbps*) of *S. suis* and updated the results. (Liu B et al., 2020; Wang Y et al., 2019; Vega et al., 2016; Gao et al., 2013; Zhang et al., 2006). Of these genes, *gor* and *gdh* were down-regulated significantly. Studies have suggested that *gor* gene is involved in the synthesis of glutathione reductase, which has been implicated in oral *L. monocytogenes* infections in mice (Mansfield and Mansfield, 2007). Glutamate dehydrogenase (GDH) is a key enzyme that connects carbon and nitrogen metabolism. It is a very important functional molecule in the energy metabolism process of bacteria, and plays a significant role in pathogenicity, which contributes to the virulence of *S. suis* type 2 (Okwumabua and Chinnapakkagari, 2005) Although it has been confirmed that AdcA and Lmb can regulate the expression of multiple genes involved in regulating bacterial virulence, the specific regulatory pathway is unknown. In the future, it would be interesting to further explore the mechanism of transcription regulation by studying the gene expression of the entire genomes of WT and knockout mutants. We have added this content to the discussion section of the article. Please refer to lines 450-463 in the revised manuscript.

(15) In Fig.5, most of the differences are below a 2-fold change. I am not convinced by the biological relevance of these differences. Moreover the title "AdcA and Lmb contribute to biofilm formation in *S. suis* by downregulating the expression of adhesion genes" is not supported by the data. There is no evidence that the difference in *perR* and *gdh* expression has any connection with the difference in biofilm formation. Additional experiments are needed to prove it.

RE: Thank you for your advice. AdcA and Lmb can not directly regulate the expression of corresponding genes. We have changed the title into "AdcA and Lmb

contribute to biofilm formation in *S. suis* by influencing bacterial virulence" in the revised manuscript. We repeated the q-RT PCR experiment and updated the results. You can refer to the question 14 for instructions.

(16) Fig. 8: Unfortunately, similar survival curves and bacterial counts in organs have been published in June 2022 by Zheng et al..

RE: Thank you for your nice suggestion. We have done animal experiment on mice before that, there are two differences between our assays. First, besides WT and $\Delta adcA\Delta lmb$, we also set $\Delta adcA$, $C\Delta adcA$, Δlmb and $C\Delta lmb$ mouse groups. At 48h after infection with 2×10^8 CFU bacteria, we detected bacterial counts. Our results provide evidence that individual *adcA* and *lmb* genes affect pathogenicity of *S. suis*, which also echoed our previous hypothesis that *adcA* may have synergistic effects with *lmb*. Besides, we detected bacterial counts in the blood at different time points (3h, 6h, 9h and 12h), which provided more detailed and comprehensive data about early stage of invasive infections of *S. suis* (Liu L et al., 2006).

(17) Line 349: remove "was" in the sentence

RE: Thank you for your nice suggestion. We have removed "was" in the sentence in the revised manuscript. Please refer to lines 355 in the revised manuscript.

(18) Line 388 "AdcA is associated with direct recruitment of Zn " : in which bacteria ? Reference ?

RE: Thank you for your nice suggestion. We have replaced this sentence with "AdcA is associated with direct recruitment of Zn in most streptococci species (Bayle et al., 2011; Makthal et al., 2017, Makthal et al., 2020)". We added the relevant references in the revised manuscript. Please refer to lines 394-398 in the revised manuscript.

(19) Line 389 "The *lmb* gene encodes a laminin, which has previously been associated with Zn acquisition". *Lmb* certainly not encodes a laminin ! It is a laminin-binding protein. Reference ?

RE: Thank you for your nice suggestion. We have replaced this sentence with "The *Lmb* is annotated as a laminin, which has previously been associated with Zn acquisition.". The reference (Tedde, V et al., 2016) has indicated this in detail. We added the relevant references in the revised manuscript. Please refer to lines 394-398 in the revised manuscript.

(20) Line 394 "The regulation mechanism of Zn transport in *S. suis* needs to be further elucidated": in numerous Streptococcal species, the Adc/*lmb* transporter is regulated by AdcR (Shafeeq et al. 2011 ; Sanson et al. 2015 ; Moulin et al. 2016). This is certainly also the case in *S. suis* and it should be discussed here.

RE: Thank you for your nice suggestion.

Group A streptococcus and *S. pneumoniae* adaptive responses to Zn limitation are coordinated by the Zn-sensing transcription regulator adherence competence repressor (AdcR) (Makthal and Kumaraswami, 2017; Manzoor et al., 2015; Sanson et al., 2015).

AdcR belongs to the multiple antibiotic resistance family of regulators (MarR) and mediates Zn-dependent transcriptional regulation of genes involved in Zn scavenging, sparing, and acquisition during Zn limitation (Reyes-Caballero et al., 2010; Sanson et al., 2015). We have added this content in the revised manuscript. Please refer to lines 221-222 in the revised manuscript.

(21) Line 417: "nutrient availability" rather than "hunger resistance"

RE: Thank you for your nice suggestion. Correction has been made as suggested in the revised manuscript. Please refer to lines 432-433 in the revised manuscript.

(22) Line 428: The authors did not provide any reference linking the tested genes with adhesion. Moreover their data do not show that the reduced biofilm formation is due to the downregulation of these genes.

RE: Thank you for your nice suggestion. We discuss this and add related reference in the discussion section of the paper. There is no evidence that AdcA and Lmb can directly regulate the expression of corresponding genes. We have changed the title into "AdcA and Lmb contribute to biofilm formation in *S. suis* by influencing bacterial virulence" in the revised manuscript. Please refer to lines 450-463 in the revised manuscript.

Reviewer #2 (Comments for the Author):

The manuscript by Mingzheng Peng et al. provides a systematic analysis of the phenotype displayed by knock-out mutants of the genes *adcA* and *lmb*, coding for the two Zn transporters in *Streptococcus suis*. The manuscript presents a useful study on the influence of zinc homeostasis in cell adhesion, invasion and biofilm formation by *S. suis* as well as on its tolerance towards antibiotics.

(1) However, a more detailed genetic characterization of the strain used should be reported and discussed as the manuscript suffers from the lack of a thorough presentation and analysis of the data. For instance, there is no mention of the source of the strain used in the study or reference to its genome sequence, which has been present in the NCBI database since 2017.

RE: Thank you for your nice suggestion. SS2 strain SC19 was isolated from a diseased pig in Sichuan, China in 2005, and the reference (Li W et al., 2009) indicated how it was obtained in detail. And we have added this information in the revised manuscript. Please refer to lines 144-145 in the revised manuscript.

(2) Furthermore, a clear comprehension of the work is hindered by the poor quality of the English language usage and punctuation. In addition, the inaccurate use of technical terminology renders the manuscript more difficult to follow.

RE: Thank you for your nice suggestion. We will continue to learn and improve our writing ability, and we will also seek help from English-speaking colleague.

(3) For example:

line 273 "complementary strains" instead of "complementation strains"

RE: Thank you for your nice suggestion. We have replaced "complementation strains" with "complementary strains" in the revised manuscript. Please refer to line 282 in the revised manuscript.

(4) line 279 strain in TSB. To explore there should be a comma instead of a full stop

RE: Thank you for your nice suggestion. We have replaced it with a comma in the revised manuscript.

(5) I strongly suggest that the manuscript should be checked by an English-speaking colleague, the clarity of the topics presented would benefit significantly.

In conclusion, the manuscript offers an extensive characterisation of the effects of loss of zinc homeostasis in *S. suis*, but the authors must capitalise better the considerable experimental work by presenting their study in a more careful manner and provide a more complete discussion of their results compared to the many relevant reports available in the literature.

RE: Thank you for your nice suggestion. We will invite relevant English-speaking colleague to polish the articles and improve the readability of articles.

February 25, 2023

Prof. Weicheng Bei
Huazhong Agricultural University
Huazhong Agricultural University
Wuhan
China

Re: Spectrum04337-22R1 (The *adcA* and *Imb* gene play an important role in drug resistance and full virulence of *Streptococcus suis*)

Dear Prof. Weicheng Bei:

Your manuscript has been accepted, and I am forwarding it to the ASM Journals Department for publication. You will be notified when your proofs are ready to be viewed.

Sincerely,

Ahmed Babiker
Editor, Microbiology Spectrum
